

# Neural network analysis of clinical variables predicts escalated care in COVID-19 patients: a retrospective study

Joyce Q. Lu, Benjamin Musheyev, Qi Peng and Tim Q. Duong

Department of Radiology, Montefiore Medical Center, Albert Einstein College of Medicine, Bronx, NY, USA

## ABSTRACT

This study sought to identify the most important clinical variables that can be used to determine which COVID-19 patients hospitalized in the general floor will need escalated care early on using neural networks (NNs). Analysis was performed on hospitalized COVID-19 patients between 7 February 2020 and 4 May 2020 in Stony Brook Hospital. Demographics, comorbidities, laboratory tests, vital signs and blood gases were collected. We compared those data obtained at the time in emergency department and the time of intensive care unit (ICU) upgrade of: (i) COVID-19 patients admitted to the general floor ($N = 1203$) vs. those directly admitted to ICU ($N = 104$), and (ii) patients not upgraded to ICU ($N = 979$) vs. those upgraded to the ICU ($N = 224$) from the general floor. A NN algorithm was used to predict ICU admission, with 80% training and 20% testing. Prediction performance used area under the curve (AUC) of the receiver operating characteristic analysis (ROC). We found that C-reactive protein, lactate dehydrogenase, creatinine, white-blood cell count, D-dimer and lymphocyte count showed temporal divergence between COVID-19 patients hospitalized in the general floor that were upgraded to ICU compared to those that were not. The NN predictive model essentially ranked the same laboratory variables to be important predictors of needing ICU care. The AUC for predicting ICU admission was $0.782 \pm 0.013$ for the test dataset. Adding vital sign and blood-gas data improved AUC ($0.822 \pm 0.018$). This work could help frontline physicians to anticipate downstream ICU need to more effectively allocate healthcare resources.

## INTRODUCTION

Since it was first reported in Wuhan, China in December 2019 (*Huang et al., 2020*; *Li et al., 2020b*; *Zhu et al., 2020b*), the coronavirus disease 2019 (COVID-19) has infected over 27 million people and killed more than 880,000 people worldwide (6 September 2020) (*Johns Hopkin University Coronavirus Resource Center, 2021*). There are recent spikes in COVID-19 cases and there will likely be second waves in many countries (*Leung et al., 2020*). To date, it is challenging for emergency room physicians to objectively and reliably determine which patients need escalated care (i.e., intensive care unit (ICU) admission) or

Corresponding author
Tim Q. Duong,
tim.duong@einsteinmed.org

anticipate ICU needs downstream for effective allocation of healthcare resources in part because much is still unknown about this disease.

Many studies have reported the use of patient demographics, clinical presentations, comorbidities, vital sign data and laboratory blood tests to predict in-hospital outcomes (see reviews (*Brown et al., 2020*; *Rodriguez-Morales et al., 2020*; *Wynants et al., 2020*)). Some earlier studies found that: (i) age and CRP thresholds are good predictor of mortality (*Lu et al., 2020*), (ii) age, lymphocyte count, lactate dehydrogenase (LDH) and $SpO_2$ are independent predictors of mortality (*Xie et al., 2020*), (iii) comorbidity, older age, lower lymphocyte and higher LDH at presentation to be independent high-risk factors for COVID-19 progression (*Ji et al., 2020*), (iv) mildly elevated alanine aminotransferase (ALT), myalgias and hemoglobin at presentation to be predictive of severe acute respiratory distress syndrome of COVID-19 with 70% to 80% accuracy (*Jiang et al., 2020*) and (v) LDH, procalcitonin (procal), $SpO_2$, smoking history and lymphocyte count were predictive of ICU admission, and heart failure, procal, LDH, chronic obstructive pulmonary disease (COPD), $SpO_2$, HR and age were predictive of mortality (*Zhao et al., 2020*). These studies have relatively small sample sizes.

Most of these published studies to predict outcomes associated COVID-19 used logistic regression. Machine learning (ML) is increasingly being used in medicine (*Deo, 2015*; *Hwang, Kesselheim & Vokinger, 2019*; *Santos et al., 2019*). ML uses computer algorithms to learn relationships amongst different data elements to relate to outcomes without the need to specify the exact relationship amongst these data elements to outcome variables. ML is ideally suited for analyzing large number of data elements. Machine learning methods outperform humans in many tasks in medicine (*Killock, 2020*). With increasing computing power and big data, ML is expected to play an important role in medicine. A neural network (NN), in particular, is an artificial NN with multiple layers between the input and output layers. NNs are based on algorithms inspired from the biological structure and functioning of a brain to aid machines with intelligence, consisting of neurons, synapses, weights, biases. A few studies have used ML to predict in-hospital outcomes associated with COVID-19 (*Hou et al., 2021*; *Li et al., 2020a*; *Yan et al., 2020*; *Yuan et al., 2020*; *Zhu et al., 2020a*).

Although some predictors of mortality and critical illnesses were shared amongst these studies, there is currently no consensus as to which clinical variables are most predictive of mortality or the needs for escalated care. Moreover, these prior studies predicted critical illness and mortality using data obtained at admission to the emergency department. COVID-19 patients came into the emergency department at various disease severity. We argue that it is more relevant to study hospitalized COVID-19 patients in the general floor who were subsequently upgraded to ICU to identify the clinical variables that predict escalated care.

Thus, the goal of this study was to identify the clinical variables that can be used to determine which patients hospitalized COVID-19 patients in the general floor will need to be upgraded to ICU early on by comparing between those not upgraded to the ICU from the general floor vs. those subsequently upgraded to the ICU. Clinical variables were obtained at the time of arrival to the emergency department and at the time of ICU

upgrade. For references, we also compared between COVID-19 patients admitted to the general floor vs. those immediately admitted to ICU. As a secondary analysis, we employed a simple neural-network algorithm to these data to identify and predict the most important clinical variables that informed the need for escalated care.

## MATERIALS AND METHODS

### Study population and data collection

This study is approved by Stony Brook University Institutional Review Board. Our IRB ID number was IRB2020-00207. Our IRB waived the need for informed consent from patients in the study. This retrospective study utilized the COVID-19 Persons Under Investigation registry ($N = 6,678$) of the Stony Brook Hospital ED from 7 February 2020 to 30 June 2020. There were 2,892 COVID-19 positive patients as determined by real-time polymerase chain reaction for severe acute respiratory syndrome coronavirus 2 (SARS-CoV-2), of which 1,430 were hospitalized. Patients who were <18 years old, still in the hospital at the time of analysis, and did not have full codes were excluded.

The final sample sizes included 1,203 patients admitted to general floor ("general floor", Group A) and 104 directly admitted to the ICU from the ED ("direct ICU", Group B), 979 patients remained on the general floor ("no upgrade", Group C) and 224 were upgraded from the general floor to the ICU ("upgrade ICU", Group D) (Fig. 1).

Demographic information, chronic comorbidities, laboratory tests, vital signs and blood gases were collected. Demographics included age, gender, ethnicity and race. Chronic comorbidities included smoking, diabetes, hypertension, asthma, COPD, coronary artery disease, heart failure, cancer, immunosuppression and chronic kidney disease. Laboratory tests included C-reactive protein (CRP), D-dimer, ferritin, LDH, white blood cell count (WBC), lymphocytes count (lymph), procal, ALT, aspartate transaminase (AST), brain natriuretic peptide (BNP), creatinine (Cr) and troponin (TNT). Vital signs included heart rate (HR), respiratory rate (RR), pulse oxygen saturation ($SpO_2$), systolic blood pressure (SBP), diastolic blood pressure (DBP) and temperature (temp). Blood gas variables and others include pH, $pO_2$, $pCO_2$, bicarbonate, sodium, hematocrit (HCRIT) and potassium.

These clinical variables were collected for general floor admission (group A) vs. direct ICU (group B) at ED admission. Data were collected for the no-upgrade vs. upgraded group at ED admission to the general floor. Data were also collected one day prior to ICU upgrade (group D) or three days after hospitalization for the no-upgrade group (Group C). The "3rd day" was chosen for comparison because the median day for patients to be upgraded to the ICU from the general floor was 3 days.

### Preprocessing and NN prediction model

Bicarbonates, $pCO_2$, $pO_2$, pH, HCRIT and TNT were not used in the ML analysis because invasive blood gas samples and TNT were not routinely obtained in our hospital on general floor patients. For the rest of the laboratory variables, missing data (<25%) were imputed using standard methods (*Van Buuren & Groothuis-Oudshoorn, 2011*).

We employed a simple NN with two fully connected dense layers using Jupyter Notebook, Tensorflow, and Keras (Fig. 2). Only two dense layers were used to avoid

Stony Brook COVID-19 Dataset Summary

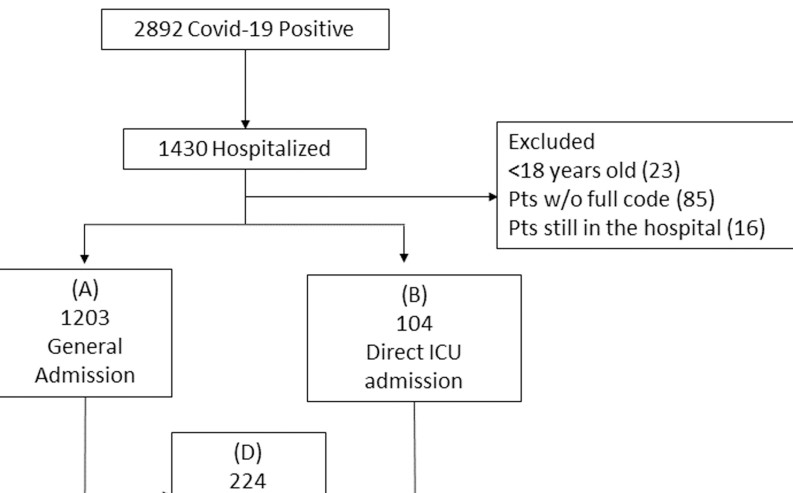

**Figure 1  Patient selection flowchart.** The final sample sizes included 1,203 patients admitted to general floor ("general floor", Group A) and 104 directly admitted to the ICU from the ED ("direct ICU", Group B), 979 patients remained on the general floor ("no upgrade", Group C) and 224 were upgraded from the general floor to the ICU ("upgrade ICU", Group D).

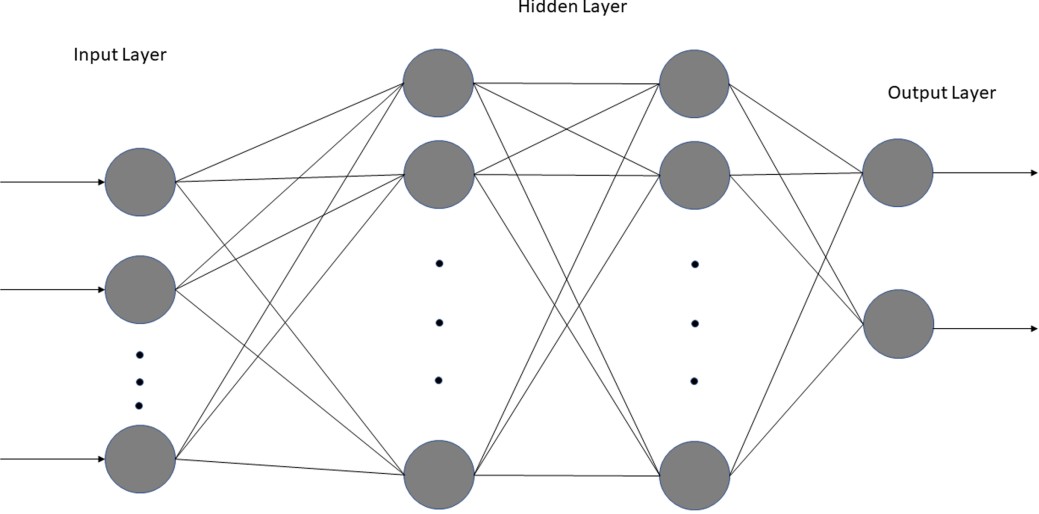

**Figure 2  Architecture of both neural networks.** The model consists of one input layer, one output layer with two fully connected hidden layers.

overfitting given large number of variables and small sample sizes. Two NN sequential models were built: one using 11 laboratory tests (excluding vitals and blood gases) and the other using laboratory tests, vitals and blood gases (total 18). The inputs consisted of the clinical variables for no-ICU vs. ICU patients: namely those of Group A (floor) at ED admission and Group C (no upgrade) at the corresponding time of upgrade vs. Group B (direct ICU) at ED admission and Group D (upgrade) at the time of upgrade. The output was ICU admission, which was binary. For both, the dataset was randomly split into 80% training data and 20% testing data. The first dense layer consisted of four nodes, and the second dense layer had three nodes, both using the rectified linear activation function (ReLU). A Softmax function for activation in the output layer was used. Training was performed for 50 epochs with a batch size of six. For the model using laboratory tests, a learning rate of 0.001 was used, whereas for the model using laboratory tests, vitals and blood gases, a learning rate of 0.0009 proved optimal. We found that these smaller learning rates resulted in the highest accuracy. The model was compiled using the Adam optimizer, an efficient gradient descent algorithm. The clinical variables were ranked using SHapley Additive exPlanations (SHAP), a Python package that explains the output of ML models based on game theory. SHAP explains the importance of a feature by calculating the contribution of each feature to the prediction. Specifically, it uses the KernelExplainer to build a weighted linear regression based on the model's predictions and the actual values from the data. It then computes Shapley values from coalitional game theory to determine the variable's importance. Lastly, we calculated cutoffs of the top 6 predictive clinical variables using a nonparametric kernel method to maximize the summation of sensitivity and specificity (*Fluss, Faraggi & Reiser, 2005*).

Statistical analysis and performance evaluation: Statistical analysis was performed using SPSS v26 (IBM, Armonk, NY, USA) and SAS v9.4 (SAS Institute, Cary, NC, USA). Group comparisons of categorical variables in frequencies and percentages were performed using the Chi-squared test or Fisher exact test. Group comparison of continuous variables in medians and interquartile ranges (IQR) used the Mann–Whitney U test. For all analyses, a $p$ value $< 0.05$ was considered to be statistically significant.

For performance evaluation of NN, data were split 80% for training and 20% for testing. Prediction performance was evaluated by area under the curve (AUC) of the receiver operating characteristic (ROC) curve for the test data set. The average ROC curve and AUC were obtained with ten runs and standard deviations were obtained. A $p$ value $< 0.05$ was taken to be statistically significant unless otherwise specified.

## RESULTS

Table 1 summarizes the demographics and comorbidities for the general floor (group A, $N = 1,203$) vs. direct ICU (group B, $N = 104$). Compared to the general floor group, the direct ICU group had more males ($p = 0.005$), smokers ($p = 0.008$), diabetics ($p = 0.047$) and patients with heart failure ($p = 0.016$). Age, ethnicity, race and prevalence of hypertension, asthmas, COPD, coronary artery disease, cancer immunosuppression and chronic kidney disease were not statistically different between groups ($p > 0.05$).

**Table 1 Laboratory tests, vital signs and blood gases of patients: (i) admitted to general floor from the emergency department ("floor") and (ii) admitted to ICU directly from emergency department ("direct ICU")**

|  | Floor N = 1203 | Direct ICU N = 104 | p Value |
|---|---|---|---|
| **Demographics** |  |  |  |
| Age, median (IQR) | 60 (49, 73) | 63 (52, 74) | 0.529 |
| Sex |  |  | **0.005** |
| Male | 687 (57.1%) | 74 (71.2%) |  |
| Female | 516 (42.9%) | 30 (28.%) |  |
| Ethnicity |  |  | 0.175 |
| Hispanic/Latino | 333 (27.7%) | 20 (19.2%) |  |
| Non-Hispanic/Latino | 710 (59%) | 69 (66.3%) |  |
| Unknown | 160 (13.3%) | 15 (14.4%) |  |
| Race |  |  | 0.784 |
| Caucasian | 629 (52.3%) | 57 (54.8%) |  |
| African American | 90 (7.5%) | 7 (6.7%) |  |
| Asian | 42 (3.5%) | 5 (4.8%) |  |
| American Indian/Alaska Native | 3 (0.2%) | 1 (1.0%) |  |
| Pacific Islander | 1 (0.1%) | 0 |  |
| More Than One Race | 7 (0.6%) | 0 |  |
| Unknown/Not Reported | 431 (35.8%) | 34 (32.7%) |  |
|  |  |  |  |
| **Comorbidities** |  |  |  |
| Smoking History |  |  | **0.008** |
| Current Smoker | 49 (4.1%) | 8 (7.7%) |  |
| Former Smoker | 250 (20.8%) | 24 (23.1%) |  |
| Never Smoker | 839 (69.7%) | 59 (56.7%) |  |
| Unknown | 64 (5.3%) | 12 (11.5%) |  |
| Diabetes | 309 (25.7%) | 36 (34.6%) | **0.047** |
| Hypertension | 573 (47.6%) | 55 (52.9%) | 0.304 |
| Asthma | 81 (6.7%) | 6 (5.8%) | 0.705 |
| COPD | 96 (8.0%) | 10 (9.6%) | 0.558 |
| Coronary artery disease | 167 (13.9%) | 15 (14.4%) | 0.878 |
| Heart failure | 84 (7.0%) | 14 (13.5%) | **0.016** |
| Cancer | 108 (9.0%) | 11 (10.6%) | 0.586 |
| Immunosuppression | 91 (7.6%) | 9 (8.7%) | 0.688 |
| Chronic kidney disease | 112 (9.3%) | 12 (11.5%) | 0.457 |

**Note:**

Group comparison of categorical variables in frequencies and percentages used $\chi^2$ test or Fisher exact tests. Group comparison of continuous variables in medians and interquartile ranges (IQR) used the Mann–Whitney $U$ test. Abbreviation: COPD, chronic obstructive pulmonary disease. IQR, interquartile range. $SpO_2$, $O_2$, oxygen saturation. Bold text indicate statistical significance.

Table 2 summarizes the demographics and comorbidities for the no-upgrade (group C, N = 979) vs. upgrade group (group D, N = 224). Compared to the no upgrade group, the upgrade ICU group had more males (p = 0.005), and patients with asthma (p = 0.008)

**Table 2 Laboratory tests, vital signs and blood gases of patients: (i) not upgraded ("no-upgrade") and (ii) upgrade to ICU from general floor ("upgrade").**

| | No-upgrade N = 979 | Upgrade N = 224 | p Value |
|---|---|---|---|
| **Demographics** | | | |
| Age, median (IQR) | 60 (49, 72) | 60 (50, 70) | 0.307 |
| Sex | | | **<0.001** |
| Male | 534 (54.5%) | 153 (68.3%) | |
| Female | 445 (45.5%) | 71 (31.7%) | |
| Ethnicity | | | 0.056 |
| Hispanic/Latino | 226 (21.2%) | 67 (29.9%) | |
| Non-Hispanic/Latino | 589 (60.2%) | 121 (54.0%) | |
| Unknown | 124 (12.8%) | 36 (16.0%) | |
| Race | | | **0.005** |
| Caucasian | 531 (54.2%) | 98 (43.8%) | |
| African American | 76 (7.8%) | 14 (6.3%) | |
| Asian | 27 (2.8%) | 15 (6.7%) | |
| American Indian/Alaska Native | 2 (0.2%) | 1 (0.5%) | |
| Pacific Islander | 1 (0.1%) | 0 | |
| More Than One Race | 7 (0.7%) | 0 | |
| Unknown/Not Reported | 335 (34.2%) | 96 (42.9%) | |
| | | | |
| **Comorbidities** | | | |
| Smoking History | | | 0.247 |
| Current Smoker | 48 (4.5%) | 5 (2.2%) | |
| Former Smoker | 237 (22.1%) | 42 (18.7%) | |
| Never Smoker | 728 (67.8%) | 164 (72.9%) | |
| Unknown | 60 (5.6%) | 14 (6.2%) | |
| Diabetes | 246 (25.1%) | 63 (28.1%) | 0.354 |
| Hypertension | 464 (47.4%) | 109 (46.7%) | 0.732 |
| Asthma | 57 (5.8%) | 24 (10.7%) | **0.008** |
| COPD | 84 (8.6%) | 12 (5.4%) | 0.108 |
| Coronary artery disease | 132 (13.5%) | 35 (15.6%) | 0.403 |
| Heart failure | 68 (7.0%) | 16 (7.1%) | 0.917 |
| Cancer | 99 (10.1%) | 9 (4.0%) | **0.004** |
| Immunosuppression | 77 (7.9%) | 14 (6.3%) | 0.410 |
| Chronic kidney disease | 94 (9.6%) | 18 (8.0%) | 0.467 |

Note:
Group comparison of categorical variables in frequencies and percentages used $\chi^2$ test or Fisher exact tests. Group comparison of continuous variables in medians and interquartile ranges (IQR) used the Mann–Whitney $U$ test. Abbreviation: COPD, chronic obstructive pulmonary disease. IQR, interquartile range. $SpO_2$, $O_2$, oxygen saturation. Bold text indicate statistical significance.

but fewer patients with cancer ($p = 0.004$). Race was different between groups. Age, ethnicity and prevalence of smoking, hypertension, diabetes, COPD, coronary artery disease, heart failure immunosuppression and chronic kidney disease were not statistically different between groups ($p > 0.05$).

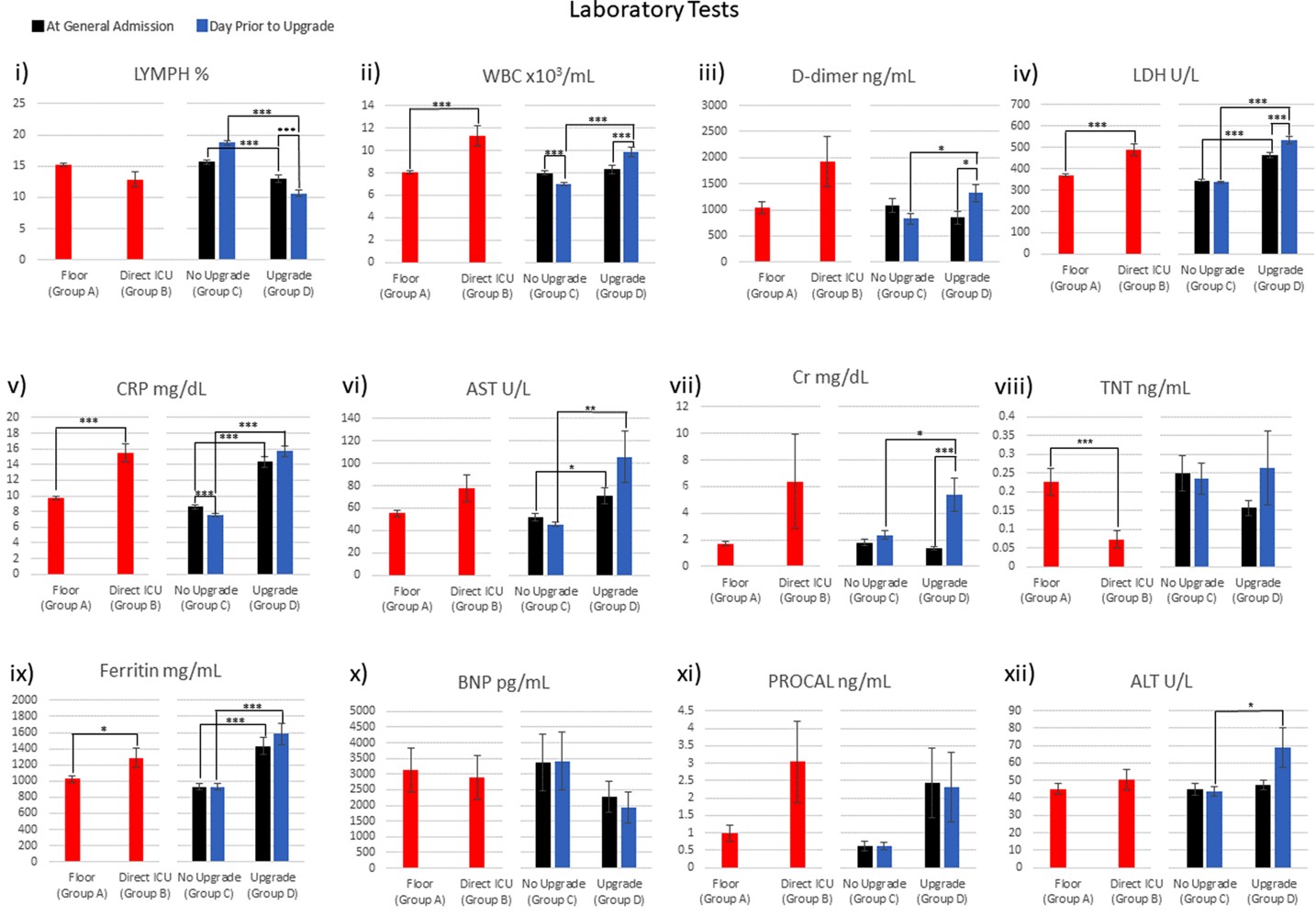

**Figure 3 Laboratory tests were collected: i) LYMPH ii) WBC iii) D-dimer iv) LDH v) CRP vi) AST vii) Cr viii) TNT ix) Ferritin x) BNP xi) PROCAL xii) ALT.** *$p < 0.05$, **$p < 0.01$, ***$p < 0.005$.

## Laboratory tests

Figure 3 plots the laboratory tests for general floor (group A) vs. direct ICU (group B) at ED admission, and no-upgrade (group C) vs. upgrade (group D) at ED admission and at the time of upgrade. WBC, LDH, CRP, TNT and ferritin were significantly different between the general floor and the direct ICU group at ED admission (red bars). Lymph, WBC, LDH, CRP, AST, CRT, ferritin and ALT were significantly different between the no-upgrade and upgrade group at the time of admission to the hospital (green bars). Lymph, WBC and CRP were significantly different between the no-upgrade and upgrade group at the day prior to upgrade (blue bars).

Table 3 integrates the comparison across different groups of Fig. 3. LDH, CRP and ferritin were significantly different for: (i) the general floor vs. direct ICU group at ED admission time point, (ii) no-upgrade vs. upgrade group at ED admission time point and (iii) no-upgrade vs. upgrade group at upgrade time point (Table 3, row 1–3). WBC stood out in that it was different for the general floor vs. direct ICU group at ED admission,
**Table 3 Comparison of laboratory tests.** This is a summary of the results in Fig. 3.

| Row | | lymh | WBC | D-dimer | LDH | CRP | Fer | Cr | TNT | AST | BNP | Procal | ALT |
|---|---|---|---|---|---|---|---|---|---|---|---|---|---|
| 1 | Group A vs. B at admission | | ↑↑↑ | | ↑↑↑ | ↑↑↑ | ↑ | | ↓↓↓ | | | | |
| 2 | Group C vs. D at admission | ↓↓↓ | | | ↑↑↑ | ↑↑↑ | ↑↑↑ | | | ↑ | | | |
| 3 | Group C vs. D at upgrade | ↓↓↓ | ↑↑↑ | ↑ | ↑↑↑ | ↑↑↑ | ↑↑↑ | ↑ | | ↑↑ | | | ↑ |
| 4 | at admission vs. at upgrade for C | | ↓↓↓ | | | ↓↓↓ | | | | | | | |
| 5 | at admission vs. at upgrade for D | ↓↓↓ | ↑↑↑ | ↑ | ↑↑↑ | | | ↑↑↑ | | | | | |
| 6 | C improved or plateau but D deteriorated | X | X | X | X | X | | X | | | | | |

**Note:**
Note that at upgrade means 1 day prior to upgrade, ↑ = significant increase where $p < 0.05$, ↑↑ = significant increase where $p < 0.01$, ↑↑↑ = significant increase where $p < 0.005$. ↓ = significant decrease where $p < 0.05$, ↓↓ = significant decrease where $p < 0.01$, ↓↓↓ = significant decrease where $p < 0.005$. X: denotes variables that showed group C improved or plateaued, but group D deteriorated between two time points.

**Table 4 Comparisons for vitals and blood gases.** This is a summary of the results in Fig. 4.

| Row | | RR | HR | SpO$_2$ | DBP | SBP | Temp | pO$_2$ | pH | pCO$_2$ | Hcrit | Blood Bicarb | Serum Bicarb | K$^+$ | Na$^+$ |
|---|---|---|---|---|---|---|---|---|---|---|---|---|---|---|---|
| 1 | Group A vs. B at admission | ↑↑↑ | | ↓↓↓ | | | ↓ | ↑↑↑ | ↓↓↓ | | | ↓ | ↓↓↓ | | |
| 2 | Group C vs. D at admission | ↑↑↑ | | ↓↓↓ | ↓↓↓ | | ↑↑↑ | | | | | | ↓↓↓ | | ↓↓ |
| 3 | Group C vs. D at upgrade | ↑↑↑ | ↑↑↑ | ↓↓↓ | | | ↑↑↑ | ↓ | ↑ | ↑ | | | | | ↓↓ |
| 4 | at admission vs. at upgrade for C | ↓↓↓ | ↓↓↓ | ↑↑↑ | ↓↓↓ | ↓↓↓ | ↓↓↓ | | | | | ↓↓↓ | ↓↓↓ | | ↑↑↑ |
| 5 | at admission vs. at upgrade for D | ↓↓↓ | ↑↑↑ | | | | ↓↓↓ | | | | | ↓↓↓ | ↓ | | ↑↑↑ |
| 6 | C improved or unchanged but D deteriorated | | | | | | | | | | | | | | |

**Note:**
Note that at upgrade means 1 day prior to upgrade, ↑ = significant increase where $p < 0.05$, ↑↑ = significant increase where $p < 0.01$, ↑↑↑ = significant increase where $p < 0.005$. ↓ = significant decrease where $p < 0.05$, ↓↓ = significant decrease where $p < 0.01$, ↓↓↓ = significant decrease where $p < 0.005$. X: denotes variables that showed C improved or plateaued, but D deteriorated between two time points.

the no-upgrade vs. upgrade at upgrade, but it was not different for the no-upgrade vs. upgrade at ED admission time point. WBC and CRP significantly decreased in the no-upgrade group (Table 4, 4th row). WBC, LDH and Cr increased while lymph decreased in the upgrade group (Table 4, 5th row).

Lymph, WBC, D-dimer, LDH, CRP and Cr improved or did not deteriorate between the two time points in the no-upgrade group but deteriorated in the upgrade group (Table 4, 6th row). These findings suggest that some of these clinical variables are informative of COVID-19 patients hospitalized in the general floor will need escalated care early on.

## Vitals and blood gases

Figure 4 plots the vital signs and blood gases for general floor vs. direct ICU at ED admission and no-upgrade vs. upgrade at ED admission and one day prior to upgrade. RR, SpO$_2$, temp, pO$_2$ and pH, were significantly different between the general floor vs. direct ICU group (red bars). RR, HR, SpO$_2$, temp, pH and pCO$_2$ were significantly different between the no-upgrade vs. upgrade group (green bars) at the time of admission to hospital. HR, SpO$_2$, DBP, SDP and temp were significantly different between the no-upgrade vs. upgrade group (blue bars) at the day prior to upgrade.

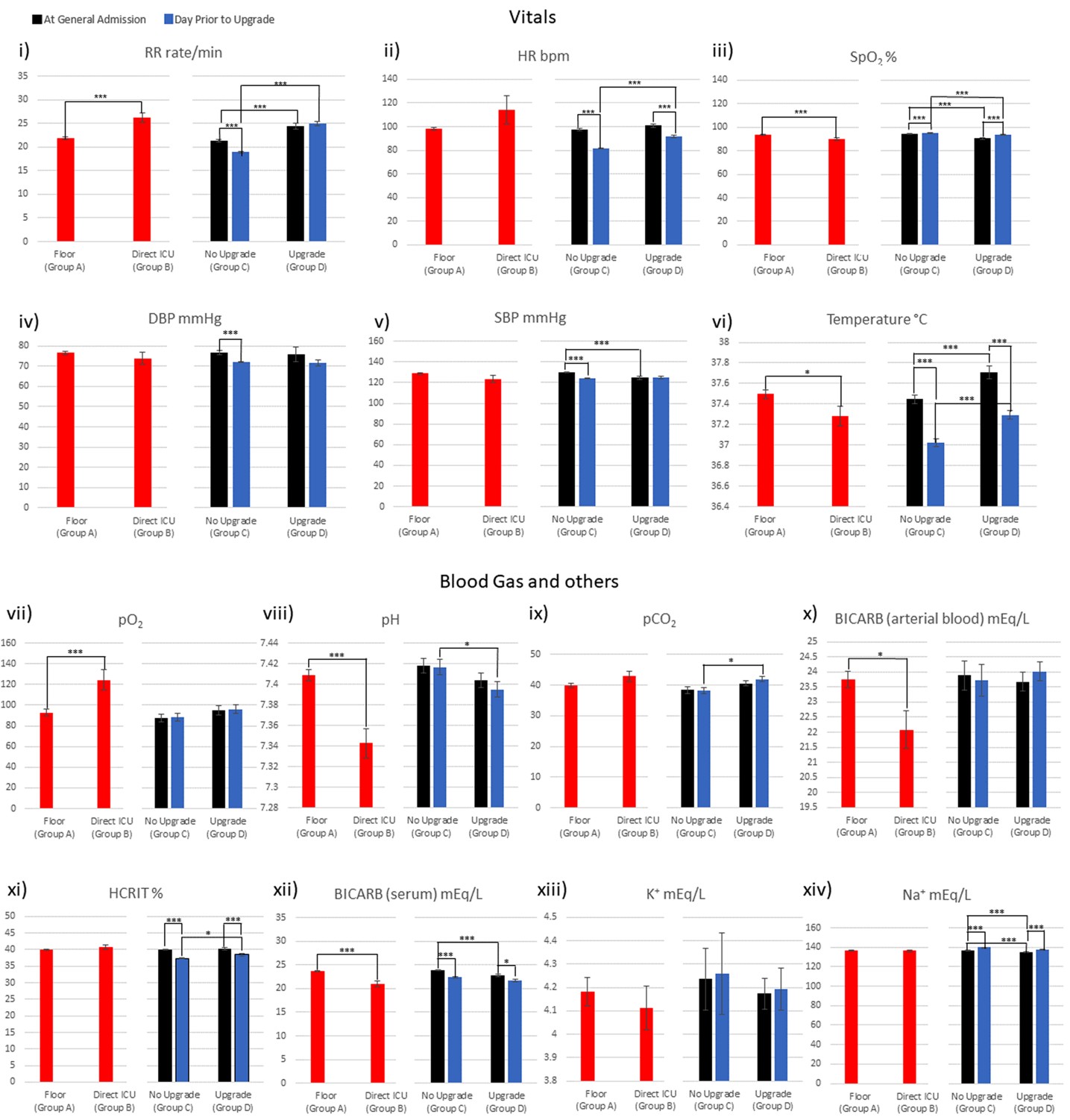

**Figure 4 Vitals, blood gases, and others were collected: i) RR ii) HR iii) SpO$_2$ iv) DBP v) SBP vi) Temperature vii) pO$_2$ viii) pH ix) pCO$_2$ x) BICARB (arterial blood) xi) HCRIT xii) BICARB (serum) xiii) K$^+$ xiv) Na$^+$.** $^*p < 0.05$, $^{**}p < 0.01$, $^{***}p < 0.005$.

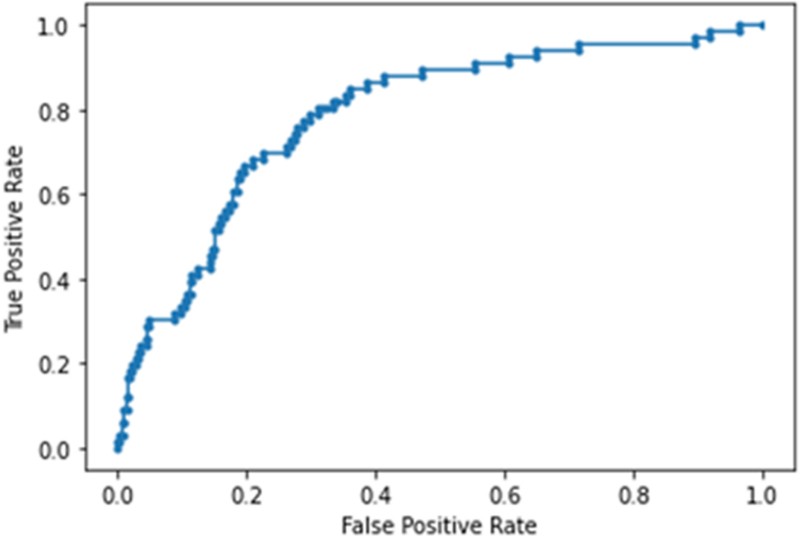

**Figure 5 AUC of deep neural network model built using laboratory tests.**

Table 4 integrates the comparison across different groups in Fig. 4. HR, SpO$_2$ and temp were significantly different for: (i) the general floor vs. direct ICU group at ED admission time point, (ii) no-upgrade vs. upgrade at ED admission time point, and (iii) no-upgrade vs. upgrade at upgrade time point (Table 4, row 1–3). pH stood out in that it was different for the general floor vs. direct ICU group at ED admission, no-upgrade vs. upgrade at upgrade but it was not different for no-upgrade vs. upgrade at ED admission.

For the no upgrade group, RR, HR, DBP, SBP significantly decreased and SpO$_2$ and temp increased (Table 4, 4th row), whereas for the upgrade group, HR and temp decreased and SpO$_2$ increased (Table 4, 5th row). Unlike the laboratory tests, none of the vitals and blood gases showed improvement in the no-upgrade group and deterioration in the upgrade group between the two time points (Table 4, 6th row). These findings suggest that some of these clinical variables are informative of COVID-19 patients hospitalized in the general floor will need escalated care early on.

## Predictors of ICU upgrade

The NN model built using laboratory tests ranked CRP, LDH, Cr, WBC, D-dimer and lymph (in order of importance) to be the top predictors of ICU admission. This model yielded an accuracy of 86 ± 5%, sensitivity of 0.242, specificity of 0.966 and AUC of 0.782 ± 0.013 for the testing dataset (Fig. 5). Note that high specificity and low sensitivity were due to sample asymmetry in which patients of ICU upgrades were fewer than those not upgraded.

The NN model built using laboratory tests, vitals and blood gases ranked RR, LDH, CRP, DBP, procal, WBC, D-dimer and O$_2$ (in order of importance) to be the top predictors of ICU admission. This model yielded an accuracy of 88 ± 7%, sensitivity of 0.364, specificity of 0.924 and an AUC of 0.822 ± 0.018 for the testing dataset (Fig. 6). Note that

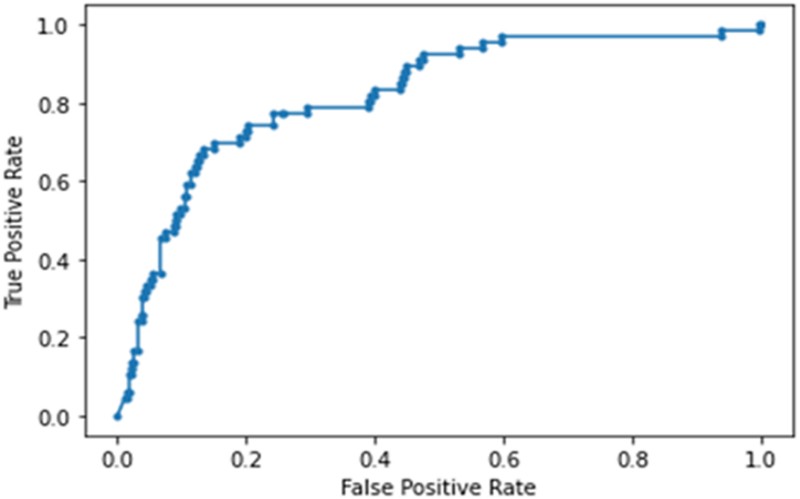

**Figure 6 AUC of deep neural network model built using laboratory tests, vitals and blood gases.**

high specificity and low sensitivity were due to sample asymmetry in which patients of ICU upgrades were fewer than those not upgraded. Cutoffs for the top predictive variables using the training set were determined to be 49.8 mg/dL for CRP, 1.05 mg/dL for Cr, 285 ng/mL for d-dimer, 392 ng/mL for LDH, 12.8% for lymph and $8 \times 10^3$/mL for WBC.

## DISCUSSION

This study investigated the clinical variables associated with direct ICU admission and upgrade to ICU from the general floor. We found that lymphocyte count, white-blood cell count, D-dimer, LDH, CRP and Cr (unranked) improved or did not deteriorate with time in patients who were not upgraded to the ICU but deteriorated in patients who were upgraded to the ICU, showing temporal divergence. The learning predictive model using laboratory tests ranked CRP, LDH, Cr, white-blood cell count, D-dimer and lymphocyte count (in orders of importance), showing substantial overlaps with those variables that exhibited temporal divergence. The performance of the predictive model using these top predictors yielded an AUC of 0.782 ± 0.013 for predicting ICU admission on the test dataset. Adding vitals and blood-gas data further improved prediction performance (0.822 ± 0.018).

Compared to the general floor group, the direct ICU group had significantly more males, smokers, diabetics and patients with heart failure. Compared to the no upgrade group, the upgrade ICU group had more males, and patients with asthma but fewer patients with cancer. Smokers, diabetics and patients with heart failure were more likely to receive escalated care at ED admission. Patients with asthma was the only comorbidity that were associated with ICU upgrade. Some major comorbidities were important factor for ICU admission especially at ED admission, but less so for ICU upgrade, suggesting that ED physicians might consider major comorbidities as factor needing escalated care.

## Clinical variables associated with ICU admission

Essentially all the laboratory test results of hospitalized COVID-19 patients were outside of normative physiologic ranges. The normative ranges of major laboratory tests were: lymphs 25–33%, WBC 0.5–11.0 × 10$^9$/L, D-dimer <250 ng/mL, LDH 45–90 U/L, CRP <10 mg/dL, AST 5–40 U/L, Cr 0.84–1.21 mg/dL, TNT <0.04 ng/mL, ferritin male: 15–200 ng/mL (male) and 12–150 ng/mL, BNP <100 pg/mL, procal <0.15 ng/mL and ALT 8–40 U/L . Elevated values of these laboratory tests indicate increased disease severity in COVID-19, except lymphocyte count where lower values are associated with worse prognosis (*Brown et al., 2020*). Note that these clinical variables could be dependent on sex, age, ethnicity and race and are shown here for reference and might not be of use clinically.

Many laboratory tests showed worse disease severity in the direct or upgrade ICU group compared to general floor and no-upgrade group. However, we found that these laboratory tests by themselves were inadequate to reliably determine which patients required ICU admission. Often time, there were no appreciable differences between those directly admitted or upgraded to the ICU and those admitted to the general floor. For example, LDH, CRP and ferritin were significantly different for the general floor vs. direct ICU group at ED admission, and no-upgrade vs. upgrade group for both ED admission and at time of the ICU upgrade, suggesting they might not be useful to distinguish ICU upgrade despite being abnormal due to COVID-19. WBC stood out in that it was different for the general floor vs. direct ICU group at ED admission and the no-upgrade vs. upgrade group at the time of upgrade, but not for the no-upgrade vs. upgrade group at ED admission, suggesting it is one of the most informative variables of ICU upgrade.

Our innovative approach was thus to identify the laboratory tests that showed improvement or plateau between the two time points in the no-upgrade group but deteriorated in the upgrade group. The laboratory tests that showed temporal divergence were identified to be *lymphocyte count, white-blood cell count, D-dimer, LDH, CRP and Cr (unranked)*. By contrast, most vitals and blood gases did not show such temporal divergence between groups, suggesting that vital signs and blood gases might be overall less important when compared to laboratory tests. This appears counter intuitive because vitals are readily available and are often informative in emergency room situation. Possible explanations are: (i) SpO$_2$ might be affected by supplemental oxygen inhalation, (ii) RR, HR, SBP and DBP could be highly variables, (iii) these vital signs were within normal normative physiological ranges (*Merck Manual for the Professional, 2020*) although there were group differences. We concluded that vital signs and blood gases appear to be overall less informative in predicting ICU admission compared to laboratory tests.

## NN analysis

To further explore whether the above-mentioned laboratory variables are predictive of direct and upgrade ICU admission, we developed a NN model, trained it on 80% of the data, and tested it independently on 20% of data that the model had not seen before. Our neural NN identified *CRP, LDH, Cr, white-blood cell count, D-dimer and*

*lymphocyte count (in orders of importance)* to be the top predictors of ICU admission. These variables showed substantial overlaps with those variables exhibiting temporal divergence described above. The performance of the predictive model using these top predictors yielded an AUC of 0.782 for predicting ICU admission from the testing dataset. Note that high specificity and low sensitivity were due to sample asymmetry in which patients of ICU upgrades were fewer than those not upgraded. Adding vital and blood-gas data improved prediction performance, yielding an AUC of 0.822 for predicting ICU admission from the test dataset. It is worth noting that RR was one of the highly ranked variables. This is not surprising because COVID-19 patients usually exhibited respiratory distress. Taken together, there is corroborative evidence that a few laboratory tests and vital signs are amongst the most important predictors of severe illness that warrants escalated care.

### Limitations

This study has several limitations. This is a retrospective study carried out in a single hospital. As in all observational studies, other residual confounders might exist that were not accounted for in our analysis. These findings need to be replicated in a large and multi-institutional setting for generalizability. It might be challenging however to achieve generalizability across hospitals because the COVID-19 pandemic circumstance is unusual and evolving and how we treat COVID-19 patient is also evolving. ICU admission rate could depend on countries, hospital practice, patient loads, available ICU beds, and when during the pandemic the data were collected, amongst others. At the time of our study, our hospital was not limited by available ICU beds. Inclusion of radiological imaging, such as chest x-ray, may be helpful in improving prediction (*Cohen et al. 2020*; *Kikkisetti et al. 2020*; *Zhu et al. 2020c*). It is conceivable that our model might not work for patients in Wuhan because their patients were more severe, amongst others. Our model also might not work on data from second COVID-19 wave. This is not because the model is wrong, but rather we believe that it is necessary to retrain predictive model with local data. We only explored NN. Alternatively, random forest, Xgboost, kernel support vector machine and other more sophisticated ML methods could also be explored. To date it is generally not trivial for hospitals to share clinical COVID-19 data because of lack of infrastructure to do so seamlessly or concerns about patient data privacy, amongst others. There is a national effort to share deidentified clinical COVID-19 data but this is not yet available.

## CONCLUSIONS

This study provided corroborative evidence that WBC, lymphocyte count, D-dimer, LDH, CRP and Cr are amongst the most important predictors of severe illness requiring ICU care. This work could help frontline physicians to better manage COVID-19 patients by anticipating downstream ICU needs to more effectively allocate healthcare resources.

### Funding

The authors received no funding for this work.

### Competing Interests

The authors declare that they have no competing interests.

### Author Contributions

- Joyce Q. Lu conceived and designed the experiments, performed the experiments, analyzed the data, prepared figures and/or tables, and approved the final draft.
- Benjamin Musheyev analyzed the data, prepared figures and/or tables, authored or reviewed drafts of the paper, and approved the final draft.
- Qi Peng analyzed the data, prepared figures and/or tables, authored or reviewed drafts of the paper, and approved the final draft.
- Tim Q. Duong conceived and designed the experiments, performed the experiments, analyzed the data, prepared figures and/or tables, authored or reviewed drafts of the paper, and approved the final draft.

### Human Ethics

The following information was supplied relating to ethical approvals (i.e., approving body and any reference numbers):

This study is approved by Stony Brook University Institutional Review Board (IRB ID: IRB2020-00207).

### Data Availability

Data is available at Kaggle: https://www.kaggle.com/justinlu24/lu-escalated-care-data

The code for the neural network and a codebook are available in the Supplemental Files.

### Supplemental Information

Supplemental information for this article can be found online at http://dx.doi.org/10.7717/peerj.11205#supplemental-information.

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
