# Peer review of "Neural network analysis of clinical variables predicts escalated care in COVID-19 patients: a retrospective study"

_PeerJ, doi:10.7717/peerj.11205_

## Round 0.1 · original submission · Major Revisions

I saw scientific merit in your work, although there are some issues which you should address in a revised version of the text.

Reviewer 1 ·

Basic reporting

-The structure of the article is completely inappropriate. The introduction should be written more appropriately.
-Related works is not provided in the introduction and be reviewed
-Descriptions of tables, pictures and diagrams should be given in the text, not in the caption.
-The quality of images and diagrams should be appropriate (at least 200 dpi).
-References should be written in journal format.

Experimental design

-The subject of the article is in the journal scope.
-Research has collected and used appropriate data and features.
-Research in the hospital and laboratory has been done well.
-The need for research in the field of the article is not well explained.
-Research on classification and the use of deep learning networks is very poor.
-Laboratory methods are well described.
-Describing and using deep learning is not appropriate at all.
-No explanation is given about the network used and how the network is configured.

Validity of the findings

-The innovation of the research is only in the use of new diagnostic features that have been well collected.
-Research in the use of deep learning lacks any innovation.
-Due to the lack of description of the deep learning network used, the presented results have no validity in the field of classification and should be reviewed in this regard.
-The results of the deep learning network are not robust.
-The results of laboratory findings are appropriate.

Additional comments

The data collected is very valuable and the research done in the laboratory is appropriate. On the other hand, the results presented in the deep learning classification are invalid due to the lack of explanation of the network configuration. It is necessary to apply the general corrections mentioned below in the article:
-The structure of the article is completely inappropriate. The introduction should be written more appropriately.
-Related works is not provided in the introduction and be reviewed.
-Avoid generalizations and the materials used should be explained realistically and in detail.
-Descriptions of tables, pictures and diagrams should be given in the text, not in the caption.
-The quality of images and diagrams should be appropriate (at least 200 dpi).
-References should be written in journal format.
-Robust deep learning network should be used.
-The deep learning network used should be described in detail and complete metrics provided.

·

Basic reporting

The structure of the manuscript is messed up and the tables and figures are not well-designed and it is hard for the reader to understand them.

Experimental design

The implemented deep model is extremely week and basic and so the achieved results are not reliable.

Validity of the findings

Not enough metrics for validation have been mentioned. In a valid evaluation, we expect the authors to mention Accuracy; Specificity; Sensitivity, and F1 score.

Additional comments

This manuscript is constructed of few fully connected layers and does not include a novelty that worth publishing in this journal.
The shared code shows the simplicity of their work. The model architecture clears this fact that the used model is the basic model of deep neural networks and has no innovation. The authors also did not mention enough metrics for evaluating their model like Accuracy; sensitivity or specificity.
For sure implementing a new deep neural network that is more advanced; will result in much higher accuracy than what the authors achieved.

---

## Round 0.2 · accepted · Accept

The manuscript has high standards to be published in PeerJ, after reading the new version, the response to the reviewers and reviewer #1's report.

Reviewer 1 ·

Basic reporting

The corrections made are appropriate.

Experimental design

The description of the classification method used (neural network) has been improved.

Validity of the findings

With the approval of experts (authors of this article), research findings can be made available to researchers.

Additional comments

Thanks for the corrections. Valuable findings have been collected that can be used in other research by improving classifications and new features.